# Characterization of Acid-Aged Biochar and Its Ammonium Adsorption in an Aqueous Solution

**DOI:** 10.3390/ma13102270

**Published:** 2020-05-14

**Authors:** Zhiwen Wang, Jie Li, Guilong Zhang, Yancai Zhi, Dianlin Yang, Xin Lai, Tianzhi Ren

**Affiliations:** 1College of Resources and Environment, Northeast Agricultural University, Harbin 150030, China; wangzhiwen612@163.com; 2Agro-Environmental Protection Institute, Ministry of Agriculture and Rural Affairs, Tianjin 300191, China; zhangguilong@caas.cn (G.Z.); zhiyancai@126.com (Y.Z.); yangdianlin@caas.cn (D.Y.); laixin@caas.cn (X.L.)

**Keywords:** biochar aging, NH_4_^+^-N adsorption, characterization, adsorption mechanism

## Abstract

According to its characteristics, biochar originating originating from biomass is accepted as a multifunctional carbon material that supports a wide range of applications. With the successfully used in reducing nitrate and adsorbing ammonium, the mechanism of biochar for nitrogen fixation in long-term brought increasing attention. However, there is a lack of analysis of the NH_4_^+^-N adsorption capacity of biochar after aging treatments. In this study, four kinds of acid and oxidation treatments were used to simulate biochar aging conditions to determine the adsorption of NH_4_^+^-N by biochar under acidic aging conditions. According to the results, acid-aged biochar demonstrated an enhanced maximum NH_4_^+^-N adsorption capacity of peanut shell biochar (PBC) from 24.58 to 123.28 mg·g^−1^ after a H_2_O_2_ modification. After the characteristic analysis, the acid aging treatments, unlike normal chemical modification methods, did not significantly change the chemical properties of the biochar, and the functional groups and chemical bonds on the biochar surface were quite similar before and after the acid aging process. The increased NH_4_^+^-N sorption ability was mainly related to physical property changes, such as increasing surface area and porosity. During the NH_4_^+^ sorption process, the N-containing functional groups on the biochar surface changed from pyrrolic nitrogen to pyridinic nitrogen, which showed that the adsorption on the surface of the aged biochar was mainly chemical adsorption due to the combination of π-π bonds in the sp^2^ hybrid orbital and a hydrogen bonding effect. Therefore, this research establishes a theoretical basis for the agricultural use of aged biochar.

## 1. Introduction

Excessive fertilization is one of the main reasons for low nutrient use efficiency, which may lead to the high risk of non-point source pollution, especially in intensive vegetable cultivation areas [1,2]. In China, the yield of commercial vegetables mostly depends on the application of a high quantity of nitrogen fertilizer. Furthermore, subsidies for artificial fertilizers from the government encourage farmers to use greater amounts of fertilizers than needed to obtain high crop output [3]. Long-term fertilization application, particularly the overuse of nitrogen fertilizers, has caused nitrogen discharge from farmland into underground water and the aquatic ecological system, which has been confirmed to be the main reason for eutrophication and has further resulted in both soil and water quality deterioration. Nitrogen leaching not only leads to the loss of soil fertility but also causes damage to the environment and human health [4,5,6]. When the rate of nitrogen fertilizer application exceeds the nitrogen requirement of a plant for a long time, it will cause the accumulation of ammonium nitrogen in the soil, thereby accelerating soil acidification. The mechanism of soil acidification caused by the application of nitrogen fertilizer is very complicated, and acidification caused by the nitrification process of ammonium nitrogen is one of the primary reasons [6,7]. The available literature mainly focuses on adsorbing nitrate and reducing nitrate leaching in biochar amended soils, but little research has focused on ammonium nitrogen adsorption. Therefore, it is necessary to improve the utilization capacity of ammonium nitrogen and reduce its loss by studying the adsorption mechanism of ammonium nitrogen.

Biochar is a carbon-rich material that originates from agricultural waste biomass and is produced by pyrolysis under oxygen-limited conditions. As a stable and attractive long-term carbon storage material, biochar has been used as an unconventional and environmentally friendly soil remediation agent to reduce pollutants from nutrients, especially nitrogen leaching, because of its highly porous structure and good adsorption capacity [8,9,10]. The characteristics of biochar are mainly determined by the raw materials and environmental factors, including temperature, heating rate, reaction time, and pressure during the preparation of biochar [11]. The specific surface area of biochar is affected by factors such as material, pyrolysis temperature, and pyrolysis method [11,12]. Generally, the specific surface area of biochar increases with the increase of cracking temperature. The pore size of biochar is related to the environmental pressure: “The larger the pressure the larger the pore size” [13]. The total number of functional groups and the density of functional groups of biochar decrease with the increase of the pyrolysis temperature and the number of acidic groups decreases while the number of basic groups increases [14]. The equipment and preparation technical conditions will also affect the characteristics of the prepared biochar. Studies have shown that when the pyrolysis temperature exceeds 400 °C, a relatively uniform microporous structure appears on the biochar surface, which is the reason why biochar has great potential for ammonium nitrogen adsorption. As the carbonization temperature increases, the isotherm adsorption curve of NH_4_^+^-N changes from linear to nonlinear. The adsorption mechanism gradually changes from partitioning to surface adsorption, thus increasing the amount of ammonium nitrogen adsorption accordingly [15]. The element composition of biochar ash mainly depends on its raw materials and pyrolysis time. Studies have found that plant-derived biochar prepared at the same temperature has a larger specific surface area than animal-derived biochar, but the ash content is significantly lower than that of animal-derived biochar [16]. The biochar surface, which is carbonized at a certain temperature, contains many oxygen-containing functional groups, such as carboxyl, carbonyl, phenolic hydroxyl, and lactone groups [11,17]. The above composition indicates that biochar has a good adsorption capacity, especially for cations. Additionally, biochar has a large number of negative charges and a high cation exchange capacity (CEC) [18], which may lead to a charge adsorption of NH_4_^+^, which has a positive charge [19]. In the research of the preparation technology improvement of biochar, it is found that the physical and chemical properties, equipment productivity, and biocarbonization rate of biochar at different carbonization temperatures can be affected by the adjustment of factors such as inlet rate and furnace pressure [20].

Fresh biochar has shown effectiveness in reducing the pollutant concentration by increasing the availability of pH-dependent cation exchange in soil, such as for heavy metal-ion pollution [21,22] and excessive nutrients [10,23]. Due to the persistence of its aromatic cluster, biochar presents stable characteristics and does not quickly decompose in soil. However, the ash of agricultural waste contains high concentrations of nutrients such as Na, Ca, and Mg, [24] and biochar with a high ash content may bring a range of inorganic compounds into soil, which may lead to more serious environmental pollution [10]. Because of the high alkalinity of an agricultural residue powder (normally between 8–13.5), fresh biochar is not suitable for application to alkaline soil. Therefore, modified biochar contained less ash and is necessary to stabilize the pH of alkaline farmland soil.

During the residence of biochar in soil, the biochar becomes oxidized by adding acidic groups to its surface over time [11,25]. Long-term application in soil may cause certain changes in the physical and chemical properties of biochar, and some of its beneficial properties may be lost over time. It is necessary to assess the long-term effects of the biochar aging process and the competitive interaction between biochar and other elements because the stability of biochar is the most decisive factor of carbon sequestration in soil [2]. Previous studies have found that natural processes, such as wind, rain erosion, freezing and thawing cycle, will change the physical structure of biochar with a long-term residence time in soil. Furthermore, the surface of biochar will gradually undergo oxidation reactions and its pH value, surface chemical composition, and non-aromatic hydrocarbon structure will change [26,27]. Kumar et al. [21] explored the physical and chemical property changes of biochar after 180-day planted plot experiments. The study showed that biochar surface oxidation occurred rapidly and significantly in the early application stage and then later remained stabilized in soil [21,28].

Artificially accelerated simulated aging can increase the oxygen content and CEC of the biochar surface, but the type of biochar and the type of pollutant have different effects on the adsorption capacity of biochar before and after aging. In farmland, due to precipitation and nitrogen fertilization, biochar will be acidified by nutrient leaching and acid rainfall after being applied in soil [29]. As a kind of common adsorbent, biochar has been widely used in soil remediation to reduce nutrient loss, especially reducing nitrate and ammonium leaching in underground water [27,30,31]. However, the adsorption effect of biochar after aging remains unknown, and previous research on aged biochar is scattered and still unsystematic. The aim of this study was to establish a theoretical reference for ammonium-N sorption behaviors by an acid and oxidation aging of biochar. In this article, peanut shells were used as the raw material, with four kinds of normal acid-treated biochar and a water-washed control group to imitate the aging and oxidization progress of biochar. Then, the adsorption principle of these acid/oxidation-treated biochar in an aqueous solution was detected.

## 2. Materials and Methods

### 2.1. Production of Biochar

Peanut shells were used as the biomass materials after smashing. The smashed shells were oven-dried at 80 °C until they reached a constant weight and sifted with a 100-mesh sieve to produce biochar by a slow pyrolysis procedure under N_2_ protection. The pyrolysis procedure followed a slow heating rate of 8 °C/min until reaching the final pyrolysis temperature of approximately 500 °C that was kept for 2 h. The peanut shell biochar was labelled PBC. The carbon content of PBC was 80.7%, with a high ash content of 16.2%, and a pH of 9.92 (carbon:water was 1:20 (g:mL). Biochar was added to deionized (DI) water and shaken at 150 rpm for 1.5 h. Then, it was allowed to stand for 10 min. The pH values were measured with a pH meter (Delta 320, Mettler Toledo, Shanghai, China) [24]. To simulate the acid and oxidization conditions in over-fertilized farmland [28], treatments with four kinds of modifiers (deionized water (18.25 MΩ/cm, produced by Millipore, Billerica, MA, USA), 2 M HCl, 2 M H_2_SO_4_, and 2 M H_2_O_2_ (Macklin, Shanghai, China) were carried out as a kind of chemical aging method to modify the biochar. The biochar modification was performed with a Soxhlet extraction apparatus in which 200 cm^3^ of liquid (the modifiers above) was added into a 250 cm^3^ round-bottom Pyrex flask with 10 g PBC and reacted at room temperature with a water condenser for 24 h. The modified biochars were washed with DI water to pH = 7 ± 0.5 and dried in an oven at 105 °C. The four modified biochars were named H_2_O-PBC, HCl-PBC, H_2_SO_4_-PBC, and H_2_O_2_-PBC.

### 2.2. Physical-Chemical Characterization Analysis of the Biochar

#### 2.2.1. Morphological, Surface, and Pore Structure Analysis

The microstructure and the surface topography of the carbon materials were characterized and analyzed by scanning electron microscopy (SEM) and energy dispersive spectroscopy (EDS). A small amount of the sample powder was spread on conductive tape, and gold was sprayed on to increase the conductivity of the carbon material. This test used a SU8100 scanning electron microscope, (Hitachi, Tokyo, Japan).

The textural properties of biochar were obtained by nitrogen and carbon dioxide adsorption/desorption tests using a specific surface tester (ASAP 2020 PLUS HD88, Mike Instruments, Norcross, GA, USA). Due to the particularity of biochar, the microporous structure of carbon material has higher adsorption selectivity towards CO_2_ than N_2_ when it has a high ash content [32]. Before testing, to reduce the adverse impact of the ash content, the test materials were degassed at 180–220 °C for over 24 h. Therefore, in this experiment, the test temperature of the CO_2_ adsorption and desorption experiment was 273 K, and the pressure range was 0–1.07 bar, which was used to test the micropore adsorption performance of the carbon materials. The specific surface area and pore volume of the micropores were calculated by density functional theory (DFT) [32]. The other pore structures were measured by the N_2_ gas method (relative gas pressure P/P0 was 0.98) and calculated by the Barrett–Joyner–Halenda (BJH) method [12].

The mineral composition of jadeite was measured by X-ray diffraction (XRD, X‘Pert-ProMPD, PANalytical, Almelo, the Netherlands). The surface charge and point of zero charge (pH_PZC_) of all adsorbents were measured through a potentiometric titration method by an automatic titrator (Mettler Toledo T70, Mettler Toledo, Shanghai, China) according to the method reported by Noh and Schwarz [33].

#### 2.2.2. Functional Groups’ Analysis

Fourier transform infrared (FTIR) spectroscopy was used to observe the change in functional groups on the various modified biochar surfaces before and after ammonium adsorption. The FTIR spectra of the samples were recorded from 4000 to 400 cm^−1^ through an FTIR spectrometer (Nicolet iS50, Thermo Fisher Scientific, Madison, WI, USA). In addition, the elemental compositions of all samples were characterized through an elemental analyzer (Vario EL cube, Elementar, Langenselbold, Germany), and C, H, O, N, and S were determined to correlate with the observed functional groups.

X-ray photoelectronic spectroscopy (XPS, ULVCA-PHI PHI 5000 VersaProbe II, Chigasaki, Kanagawa, Japan) was used to determine the elemental compositions and functional groups on the surface of all samples. The core level spectra were analyzed using CasaXPS 2.3 software (Version 2.3.15, CasaXPS software Ltd., Manchester, UK).

### 2.3. Batch Sorption Experiments

(NH_4_)_2_SO_4_ with a concentration of 1000 mgN/L was used as a stock solution and stored in a 4 °C refrigerator; in the adsorption test, the stock solution was diluted with deionized water. All samples were filtered (0.45-μm filter) to ensure no interference effects for the NH_4_^+^ molecule and then analyzed by a continuous flow chemical analyzer (AA3, Bran+Luebbe Corp, Hamburg, Germany). According to the preliminary experiments, the results showed that at pH values between 6–8, PBC with a mass from 0.1 to 1 g could remove up to nearly 80% NH_4_^+^-N from a biochar-nutrient mixture solution with 1000 mg/L NH_4_^+^. PBC values greater than 1 g showed no additional adsorption after 20 h of shaking, thus, the adsorbed amount reached the maximum. Therefore, subsequent batch sorption experiments were investigated at pH = 7 ± 0.5 in which 100 mg of adsorbent was added to 50 mL of absorption solution (with a concentration of 20 to 1000 mg N/L) and shaken at 200 rpm for 24 h.

The equilibrium adsorption amount of NH_4_^+^-N per unit mass of sorbent was calculated by the following equation:(1)$$Q_{e}=\frac{(C_{0}-C_{e})*V}{m}\;$$

C_0_ and C_e_ (mg/L) are the initial and equilibrium concentrations of NH_4_^+^-N in solution, respectively, Q_e_ (mg/g) is the adsorbance of NH_4_^+^-N at equilibrium, V (L) is the volume of the solution, and m (g) is the mass of biochar.

A concentration of 100 mg N/L (NH_4_)_2_SO_4_ solution was selected for investigating the adsorption kinetics at room temperature (25 °C) under pH = 7 ± 0.5. Due to the adsorption conditions, 100 mg biochar was added to 50 mL adsorbent, and the samples were measured after 5, 10, 20, 30, 60, 90, 120, 180, 360, 720, 1080, and 1440 min at 200 rpm.

The equilibrium adsorption amount of NH_4_^+^-N per unit mass of sorbent was calculated using the following equation:(2)$$Q_{t}=\frac{(C_{0}-C_{t})*V}{m}$$

Q_t_ (mg/g) is the amount of NH_4_^+^ sorbed at a given time interval (t). C_0_ (mg/L) is the initial NH_4_^+^ concentration. C_t_ (mg/L) is the NH_4_^+^ concentration in solution at time t. V (L) is the volume of solution and m (g) is the mass of biochar.

### 2.4. Modeling the Adsorption Data

According to previous studies of adsorption isotherms, it is convenient to optimize the use of adsorbents by modelling the interaction between adsorbates and adsorbent materials during the adsorption process [11,34]. Therefore, adsorption kinetics were used to discover the mechanism of the thermodynamic and dynamic properties of ammonium-ion adsorption on biochar [35]. In this article, Langmuir, Freundlich, pseudo-first-order, and pseudo-second-order kinetics models and the Webber–Morris (diffusion) equation [36,37] were used to evaluate the ammonium-ion sorption phenomenon on the different acid-modified biochar samples.

#### 2.4.1. Adsorption Isotherms

The Langmuir equation can be expressed as:(3)$$Q_{e}=\frac{{\mathrm{bC}}_{e}Q_{m}}{1+{\mathrm{bC}}_{e}}$$

The Freundlich equation can be expressed as:(4)$$Q_{e}=K_{F}{C_{e}}^{1/n}$$
where Q_e_ (mg·g^−1^) is the amount of solute adsorbed per unit weight of sorbent, C_e_ (mg·L^−1^) is the equilibrium concentration of NH_4_^+^ in the bulk solution after adsorption, Q_m_ (mg·g^−1^) is the monolayer adsorption capacity, b (mg·L^−1^) is the constant related to the affinity between the adsorbate and adsorbent, K_F_ (mg^(1 − n)^·L^n^·g^−1^) is the constant related to the relative adsorption capacity of the adsorbent, and 1/n is the constant related to the adsorption intensity [38,39].

#### 2.4.2. Adsorption Kinetics

For the pseudo-first-order model (Lagergren model) and pseudo-second-order model, the following expressions were used [40]:(5)$$Q_{t}=Q_{e}\left(1-e^{\frac{-k_{1}t}{2.303}}\;\right)$$
(6)$$\frac{t}{Q_{t}}=\frac{1}{k_{2}{Q_{e}}^{2}}+\frac{t}{Q_{e}}=\frac{1}{v_{0}}+\frac{t}{Q_{e}}$$
where Q_e_ (mg·g^−1^) is the adsorbed amount of NH_4_^+^ at equilibrium, Q_t_ (mg·g^−1^) is the amount of NH_4_^+^ sorbed at time t, k_1_ (min^−1^) and k_2_ (min^−1^) are the rate constants of pseudo-first-order and pseudo-second-order adsorption, respectively, and v_0_ (g·mg^−1^·min^−1^) is the initial adsorption rate.

The following equation was used for the intraparticle model [41]:(7)$$Q_{t}={K_{d}}^{1/2}+c$$
where Q_t_ (mg·g^−1^) is the amount of NH_4_^+^ sorbed at time t, K_d_ (mg/g·min^1/2^) is the intraparticle diffusion rate constant and c (mg·g^−1^) is the boundary layer diffusion constant. The graphical plots of the linear relationship between Q_t_ and t^1/2^ show K_d_ as the slope and c as the intercept.

## 3. Results and Discussion

### 3.1. Characterization of Biochar

#### 3.1.1. Specific Surface Area, Pore Structure, and Elemental Content Analysis

According to the results, aging treatments could greatly enhance the fixed carbon of peanut shell biochar from 42.36% to 71.52% (Table 1). Moreover, the ash contents of PBC decreased to 10% from over 30% after acid and oxidation aging treatments. Most ash contents exist as soluble salts, which are easily dissolved in water [42], and the ash removal efficiency of biochar under an oxidation treatment (H_2_O_2_) is higher than that under an acid treatment and water control group. The main reason for the high pH of biochar is the alkaline substances that form by the mineral elements of ash dissolving in water [14,43]. The pH of the zero point of PBC (pH_PZC_) was reduced by removing the ash from the biochar surface. Due to the particularity of biochar, the microporous structure of the carbon material has a higher adsorption selectivity toward CO_2_ than N_2_ when it has a high ash content [44]. Therefore, the surface area and pore volume of the micropores were measured with the CO_2_ gas method and calculated by the DFT method, while the other pore structures were measured by the N_2_ gas method and calculated by the BJH method [41].

In this study, all of the biochar samples had highly microporous structures, and the surface area and pore volume of the micropores were much higher than those of the mesopores and macropores (Figure 1 and Appendix A). Peanut shell biochar materials do not contain only micropores of uniform size. Instead, the biochar materials contain a large number of irregular mesopores and macropore structures that are distributed in addition to micropores. In general, the acid and oxidation treatments could increase the surface area and pore volume of biochar pore structures, especially for the biochar micropores. However, in the biochar samples treated with sulfuric acid and hydrochloric acid, the surface area and pore volume of the mesoporous and macroporous structure have been decreased because the mesopores and macropores were destroyed by acid treatment. Meanwhile, the microporous structures of biochar were barely changed under the overly acidic treatment process. This result is in accordance with most research results [12,45]. Tang and Zhu mentioned that carbon-rich materials with highly porous states effectively capture nitrogen and phosphate ions and exhibit fast adsorption rates [23,46]. Therefore, compared with the original biochar, the acid- and oxidation-aged biochar should have more advantages in the adsorption process.

The SEM surface morphology analysis of the PBC and water- and acid-washed biochars is illustrated in Figure 2A–E, and it is difficult to clearly observe the pore structure on the surface of unmodified PBC. Combined with the results in Table 1, the PBC surface was covered with a large amount of ash, and the PBC structure was mostly irregular and amorphous (Figure 2A). Water washing biochar (H_2_O-PBC) could significantly reduce the ash content and pyrolysis by-products on the PBC surface so that some parts of the original porous structure of the biochar could be exposed (Figure 2B). Although the 2 M hydrochloric acid and sulfuric acid treatments could effectively remove the ash on the biochar surface (Table 1), the SEM images showed that the HCl and H_2_SO_4_ treatment could cause erosion of the biochar material (Figure 2C,D). Furthermore, although the specific surface area of the micropores did not change much, the original and regular pore structures disappeared after the treatment (Figure 2C,D). In contrast, as a kind of weakly acidic oxidant, hydrogen peroxide not only had a good effect on removing ash content from the biochar surface but also did not cause significant damage to the pore channel structure. Compared with that after the water treatment, the inside of the pore was cleaned more thoroughly and the pore surface was smoother after the hydrogen peroxide treatment (Figure 2E).

The elemental analysis results (Appendix A) of the biochar showed that the C content in the five kinds of biochar all exceeded 60%, which was higher than that of general commercial biochar [16]. In previous research, acid washing, especially strongly oxidizing acid washing, can import oxygen-containing functional groups to biochar, thereby increasing the percentage of H/C and O/C [12,28]. The relative elemental content ratio (H/C, O/C) of biochar can be used to characterize the aromaticity, hydrophilicity, and polarity of carbon materials. Among them, the smaller H/C is the stronger the aromaticity and hydrophilicity of the material; in contrast, the higher the O/C value is the greater the polarity of biochar [47]. The results of Appendix A show that, compared with the unmodified PBC, the aromaticity and hydrophilicity of the modified PBC increased slightly and the polarity decreased overall.

#### 3.1.2. Surface Functional Group Analysis and X-ray Crystal Structure Analysis

An XRD pattern analysis of biochar (Figure 3) shows that there are significant bulges in the diffraction pattern from 15° to 30° before and after the acidification of PBC, and this region represents amorphous organic components [20], indicating that the amorphous organic components on the biochar surface did not change significantly before and after the water and acid treatments. The surface of both the PBC- and acid-treated biochar samples contained a few inorganic components, such as silicon oxide and calcium and magnesium carbonates. Compared with PBC, the acid-treated biochar samples exhibited a more stable crystal structure in which the inorganic substances changed to more stable crystals and their oxides. While some studies have found that different crystal components indicate the different biomaterials in biochar [19,21], however, KCl, SiO_2_ and aluminium oxide hydroxide crystals often exist in straw and peanut shell biochar [48]. Furthermore, an acid treatment can stabilize the chemical structure of the biochar carbon skeleton [49], and the stable structure of carbon-based materials often represents better adsorption capability of nutrients from wastewater [37], which is consistent with the results in this research.

Figure 4a shows the infrared spectra of the five kinds of biochar. As shown in the figure, the infrared absorption peaks of the five kinds of biochar before and after the aging modification are roughly similar. The absorption peak at 3400 cm^−1^ represents the associated hydroxyl (O–H) stretching vibration peak or water molecules in the sample. Carbohydrates are the main source of the hydroxyl group in the biomass. Moreover, the absorption peak at 1628 cm^−1^ is an asymmetric stretching vibration of a C–O or C=O group on a benzene ring. The absorption peak at 1383 cm-1 corresponds to the in-plane flexural vibration of the aromatic –CH_3_ group on a benzene ring, and 910–650 cm^−1^ is the benzene ring substitution region, which represents the out-of-plane bending vibration of a C–H bond on the benzene ring. The infrared analysis results show that the oxidative modification in this paper did not change the functional group of the biochar. Although the absorption peak of the hydroxyl group changed to some extent, it may be due to the interference of water molecules.

The FTIR analysis of NH_4_^+^-N adsorbed PBC (Figure 4) illustrated that the main absorption peaks of the FTIR spectrum before and after adsorption changed greatly, and the changed absorption peaks mainly appeared at 3430, 2000, 1628, and 1383 cm^−1^, which corresponded to the characteristic peaks of the stretching vibration of the hydroxyl, carboxyl, and amide groups on the biochar surface, respectively [17]. Therefore, the sorption of NH_4_^+^-N by biochar could be mainly caused by the hydroxyl and carboxyl groups on the carbon surface, which could form coordination bonds with the nitrogen atom. At the same time, in the infrared spectrum, the absorption peaks between 1000 and 650 cm^−1^ are the combined vibration of benzene ring substituents on the biochar surface. The dramatic change before adsorption showed that NH_4_^+^-N sorption by the biochar occurred through electrostatic induction or hydrogen bonding on the carbon surface, and nitrogen atoms were adsorbed to form amino groups in a manner that the alkyl group on the benzene ring structure was substituted.

To further determine the changes in functional groups and chemical bonds on the biochar surface, XPS was used to verify the chemical composition and the presence of elements. From the full-spectrum XPS scan in Appendix A, the PBC and aged PBCs were mainly composed of carbon (combined energy of approximately 285 eV), oxygen (combined energy of approximately 400 eV), and oxygen (combined energy of approximately 530 eV) [14]. Through a peak-differentiation-imitating analysis of C 1s, the C of aged biochar can be clearly divided into four forms. The characteristic peak at 284.5 eV corresponded to the functional groups CC, CH, and C=C hybridized to sp^2^ and sp^3^. The C bond in the form of a double bond mainly represents the graphitized structure of the carbon material, and the C of the single bond represents the helical structure of the aromatic carbon and benzene rings. The characteristic peak at 285.4 eV corresponds to the C–O functional group belonging to alcohols and phenols, the characteristic peak at 286.4 eV corresponds to the C=O functional group in the carbonyl group, and the characteristic peak at 289 eV corresponds to the O–C=O functional group in the ester group.

The percentage of oxygen-containing functional groups in the peak area was calculated to obtain semi-quantitative analysis results of various functional groups on the biochar surface (Appendix A). In general, the biochar surface contains massive oxygen-containing functional groups, and oxidative modification increased the percentage of oxygen-containing functional groups on the biochar surface in all C-containing functional groups. However, the analysis of the oxygen spectrum found that there was no specific advantageous O functional group, and the oxidative modification did not significantly increase the O–H functional group content on the biochar surface. This result was mutually verified with the FTIR analysis results above, and it could be indicated that the change in the hydroxyl (O–H) stretching vibration peak in the infrared spectrum was mainly affected by the water molecules in the biochar. The presence of a large number of oxygen-containing functional groups on the biochar surface (C–O, C=O, and O–H) could not only improve the hydrophilicity of the material on the biochar surface but also increase the variety of charge on the aged biochar surface, thus enhancing their ability to adsorb ions in water and soil.

According to the peak-separation results before and after NH_4_^+^-N adsorption, the calculation results of the percentage of N-containing functional groups showed that they changed greatly on the PBC surface after adsorption (Figure 5). More specifically, the PBC and aged PBCs contained two kinds of N-functional groups, pyridinic-N and pyrrolic-N, respectively, and most of the N-functional groups existed in the form of pyrrole before N adsorption. The main N-containing functional groups on the PBC surface changed from pyrrolic nitrogen to pyridinic nitrogen, and there were still a small number of nitrate N groups on the surface of H_2_SO_4_-PBC. The nitrogen content of the aged biochar increased significantly after NH_4_^+^-N adsorption. This shows that the NH_4_^+^-N adsorption mechanism of the oxidation-aged biochar was mainly chemical adsorption, and the π-π bond in the sp^2^ hybrid orbital participated in the adsorption process.

In contrast to the variation in peak position and peak area of C and O (Appendix A), the peak positions showed no significant change, but the peak area of O-containing functional groups decreased dramatically after NH_4_^+^-N adsorption. Thus, it could be proven that the adsorption processes of NH_4_^+^-N on acid-aged biochar consumed the oxygen-containing functional groups on the biochar surface, which combined with the chemical bonds, hydrogen bonds, and π-π bonds through the chemical adsorption process. This result agrees with the EDS analysis results (Appendix A), and the N distribution areas on the biochar surface greatly coincide with the O areas after NH_4_^+^-N adsorption. The whole adsorption process of acid-aged biochar also included physical adsorption, including ionic interactions.

### 3.2. Adsorption Isotherm

The adsorption data were fitted with the Langmuir and Freundlich models for studying the NH_4_^+^-N sorption isotherms on the biochar. The fitting curves are clarified in Figure 6, and the corresponding parameters are listed in Table 2. Depending on the correlation coefficient (R^2^), all sorption behaviours of NH_4_^+^-N on the five kinds of biochar were fitted better by the Langmuir equation (R_L_^2^ = 0.986~0.993) than by the Freundlich equation (R_F_^2^ = 0.934~0.988). On the theoretical basis of the Langmuir model, which represents monolayer adsorption, better fitting results suggested that NH_4_^+^-N adsorption was proceeded by monolayer chemisorption within the adsorbents [37]. The maximum capacity of NH_4_^+^-N sorption with H_2_O_2_-PBC could be calculated as 123.227 mg·g^−1^ (Q_m_), which was a great improvement on the sorption ability of PBC for NH_4_^+^-N (24.575). Compared to other relevant study results of ammonium sorption, most biomass-based biochar cannot provide high uptake capacity in acidic or neutral environments [16]. More specifically, the maximum ammonium removal efficiency of biomass biochar and agricultural residues, such as rice husks or plant shells, are below 40% [37,50]. Although active modifications by metal ions or acids and bases can increase the NH_4_^+^ sorption capacity to more than 100 mg·g^−1^ [51], biochar after such an active process is not suitable for agricultural applications to avoid further environmental impact [52]. Moreover, the Freundlich equation also fit well with the sorption data, particularly for the water and acidified biochar (R^2^ > 0.95). The results indicated that the biochar surface was heterogeneous and that some level of physical adsorption occurred on the surface [50]. The values of constant 0 < 1/n < 1 implied that the sorption process was inclined to both a chemical and slightly physical adsorption [23,25].

### 3.3. Adsorption Kinetics

Figure 7 shows the sorption kinetics of ammonium on PBC and PBC after water and acid washing. The parameters are listed in Table 3. As plotted in Figure 7, NH_4_^+^-N adsorption of selected adsorbents reached the apparent sorption equilibrium point after approximately 180 min. According to the regression coefficient (R^2^), both pseudo-first-order and pseudo-second-order models fit well with the adsorption data. The amount of ammonium adsorbed on acidified PBC was much higher than that adsorbed on the water-washed and original PBC. The biochar after an acid treatment displayed a higher initial sorption rate (v_0_) than the water-washed and original PBC (Table 3), and H_2_O_2_-PBC showed the highest v_0_ (1.978 mg/g/min) among the five adsorbents. Moreover, the adsorption would reach the equilibrium point after approximately 180 min in a 100 mg/L NH_4_^+^-N solution. Based on the fitting results, the rapid sorption on the selected adsorbents might be regulated by chemical interactions. From previous studies [40,50], the adsorption velocity on granular porous adsorbents was controlled by intraparticle diffusion.

To identify the effects of intraparticle diffusion and boundary diffusion during the adsorption process, the fitting results of the intraparticle diffusion model are shown in Figure 8. The fitting curves of the acidified biochar samples could be linearly fitted to three stages, and the adsorption process of ammonium-N on PBC and H_2_O-PBC could be linearly fitted to two stages. More specifically, the straight line fitting of the first stage could represent the relation between intraparticle diffusion adsorption and the adsorption rate, the fitting line of the second stage implied the boundary layer diffusion effect during the adsorption process, and the last stage was the final sorption equilibrium stage [40,53]. For the whole adsorption process of PBC and H_2_O-PBC, except in the equilibrium stage, the fitting line nearly passed through the origin in the first stage, implying that intraparticle diffusion adsorption was the key controller of the adsorption rate. For the three kinds of acidified biochar, the fitting line almost passed through the origin in the first stage, demonstrating that the effect of intraparticle diffusion was one of the important rate-controlling aspects for NH_4_^+^-N adsorption, while the high positive value of boundary layer diffusion (c) in the second stage showed that apart from intraparticle diffusion, the adsorption rate of NH_4_^+^-N on the selected acidified biochar might also be affected by boundary layer diffusion [54].

Both the equilibrium and kinetics data showed that the major NH_4_^+^-N sorption mechanism on the treated biochar in this article was chemisorption, and the SEM and Brunauer-Emmett-Teller (BET) analysis results suggested that the physical interaction caused by porous adsorption was not the primary reason for NH_4_^+^-N removal from an aqueous solution, which was similar to the results of Wang’s research, NH_4_^+^ adsorption on biochar is mainly related to acidic functional groups on biochar surface [55]. According to the SEM-EDS analysis (Appendix A) after NH_4_^+^ adsorption, the elemental distribution map showed that N increased after adsorption, and the N distribution was quite similar to the site of the oxygen atom. Combined with the FTIR spectra in Figure 4, the nitrogen atoms were coordinated with the carboxyl and hydroxyl groups on the biochar surface through the spectral observation of the hydroxyl, carboxyl, and amide group stretching vibration changes at 3430, 2000, 1628, and 1383 cm^−1^. The spectra between 1000 and 650 cm^−1^ were the binding vibrations of the substituent groups of benzene, which could be used to prove the substitution of alkyl groups by amino groups on the benzene structure via inductive effects or hydrogen bonds [14].

Table 4 compares the maximum nitrogen adsorption capacity of the biochar used in this study with that of the carbon materials used previously. Generally, the maximum NH_4_^+^-N adsorption amount of agricultural biomass biochar was between 10–50 mg·g^−1^, when the PBC is at an intermediate level for ammonium adsorption. In this research, the four kinds of aging biochar had better adsorption performance, especially H_2_O_2_-PBC had higher adsorption performance than most other adsorbents. However, activated biochar will normally have better NH_4_^+^-N sorption capacity than other carbon materials.

## 4. Conclusions

In this article, four kinds of acid and oxidation treatments were used to simulate the aging process of biochar under acidic conditions in farmland with untreated fresh biochar as the control group. Compared with the water-washed and untreated biochar, acid-washed biochar can greatly enhance the adsorption ability of NH_4_^+^-N in an aqueous solution. Hydrogen peroxide-aged biochar had the highest amount of NH_4_^+^-N (123.23 mg·g^−1^) sorption among the five kinds of prepared biochar. Compared with other similar research on NH_4_^+^ adsorption, the biochar in this study provided a relatively high NH_4_^+^ sorption capacity. The surface morphology, material composition, and sorption capacity of the biochar changed after the aging treatment. As a result of redox reactions, the oxygen-containing groups on the biochar surface increased, the inorganic minerals were washed off, the polarity increased, and the aromaticity decreased. With the removal of the ash content on the biochar surface by an acid aging treatment, the binding points of inorganic cations were exposed, and the mesoporous and macroporous volume of the biochar was enlarged, which was conducive to NH_4_^+^-N adsorption. In addition, the acid treatment destroyed the microporous surface structure of the biochar; so as the adsorption site was destroyed, the adsorption capacity was weaker than that of the oxidized treatment. According to the adsorption isotherm models and kinetic models, NH_4_^+^-N sorption on the aged biochar in this study was mainly related to oxygen functional groups on the biochar surface. The improved NH_4_^+^-N adsorption ability might be related to the chemisorption and porous interparticle diffusion of ammonium atoms on the biochar. Therefore, it could be said that the acid and oxidation aging treatments did not decrease the adsorption ability of PBC in this simulation experiment. Moreover, aged biochar under a weakly acidic oxidant (H_2_O_2_) could highly improve its adsorption ability towards NH_4_^+^-N. The further influences of the aging treatment on the biochar adsorption process need to be further investigated through field tests. In the experiment of simulating the natural aging process, this paper only considered the method of simulating chemical oxidation, and further research needs to be conducted under soil conditions.

## Figures and Tables

**Figure 1 materials-13-02270-f001:**
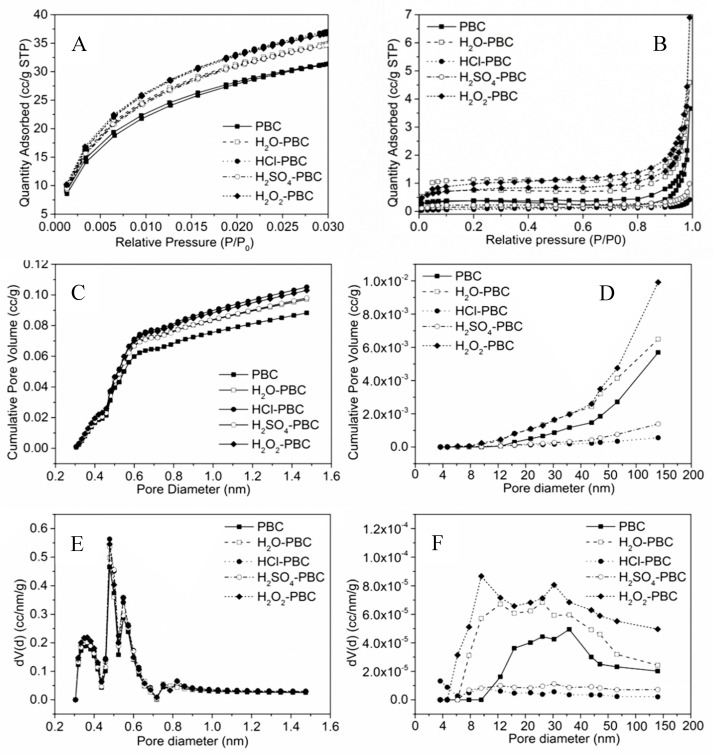
The gas adsorption, desorption isotherms and the distribution curves of the biochar pore size. ((**A**,**C**,**E**) were adsorbed by CO_2_ and (**B**,**D**,**F**) were adsorbed by N_2_).

**Figure 2 materials-13-02270-f002:**
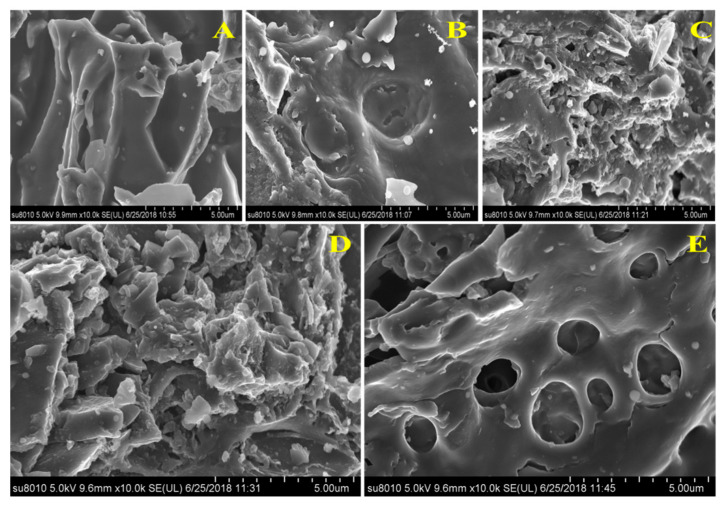
Scanning electron microscopy images of the five biochar samples. (**A**–**E**) represent the images for PBC, H_2_O-PBC, HCl-PBC, H_2_SO_4_-PBC and H_2_O_2_-PBC, respectively.

**Figure 3 materials-13-02270-f003:**
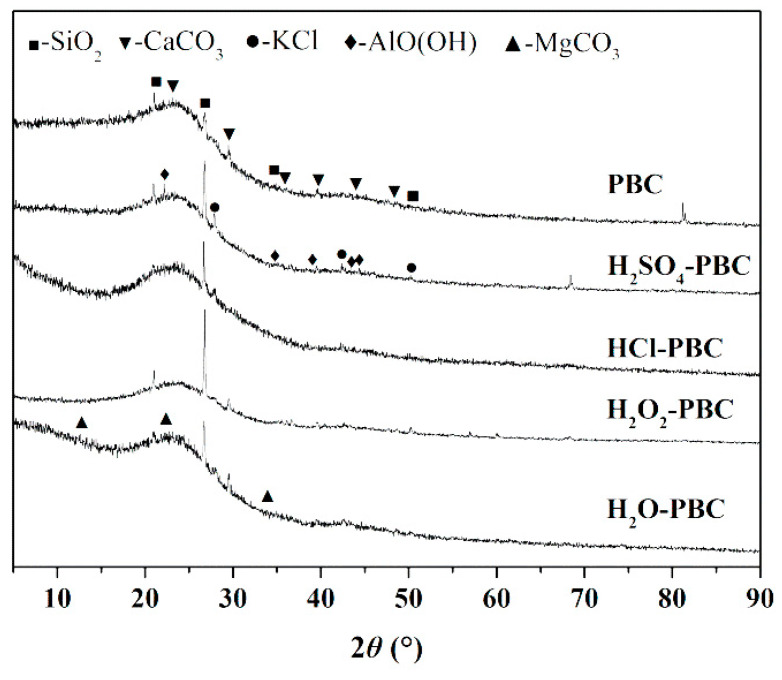
XRD patterns of the different biochar samples.

**Figure 4 materials-13-02270-f004:**
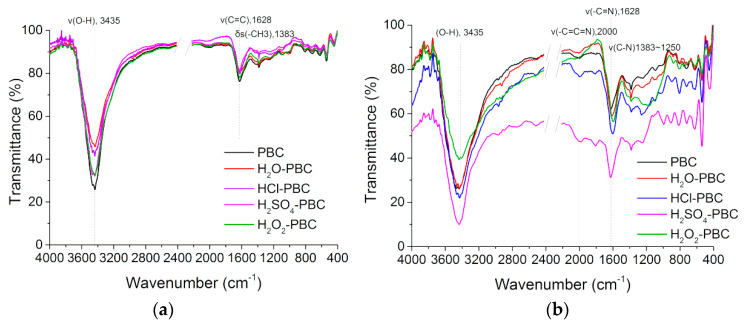
FTIR spectra of the biochar samples before (**a**) and after the adsorption of NH_4_^+^-N (**b**).

**Figure 5 materials-13-02270-f005:**
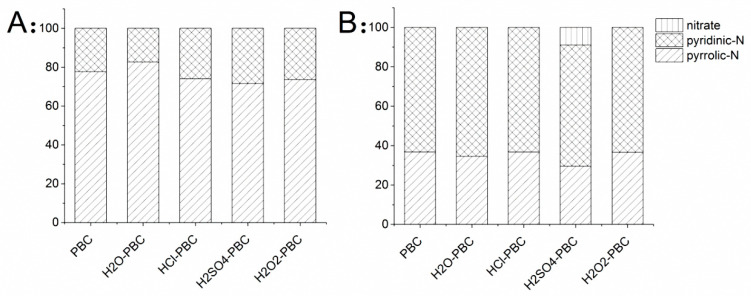
The ratio of N-functional groups on the biochar before and after NH_4_^+^-N adsorption (**A**: before adsorption and **B**: after adsorption).

**Figure 6 materials-13-02270-f006:**
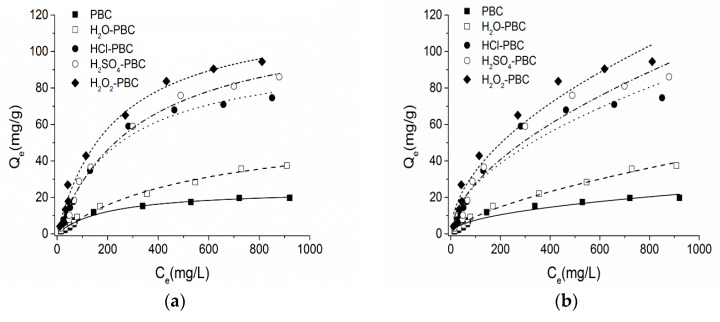
Adsorption isotherms of NH_4_^+^-N on the biochar samples as well as modeling with the Langmuir (**a**) and Freundlich (**b**) models.

**Figure 7 materials-13-02270-f007:**
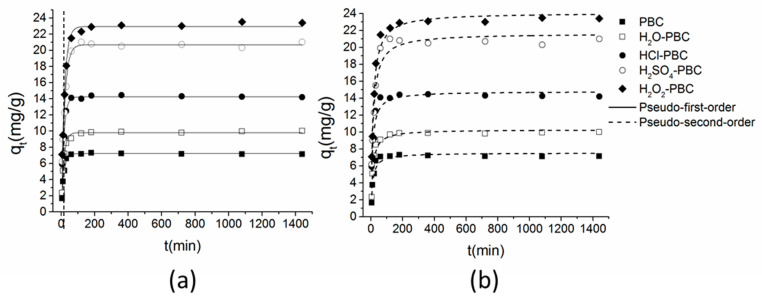
Adsorption kinetics of NH_4_^+^-N on the five kinds of biochar fitted with the pseudo-first-order (**a**) and pseudo-second-order models (**b**).

**Figure 8 materials-13-02270-f008:**
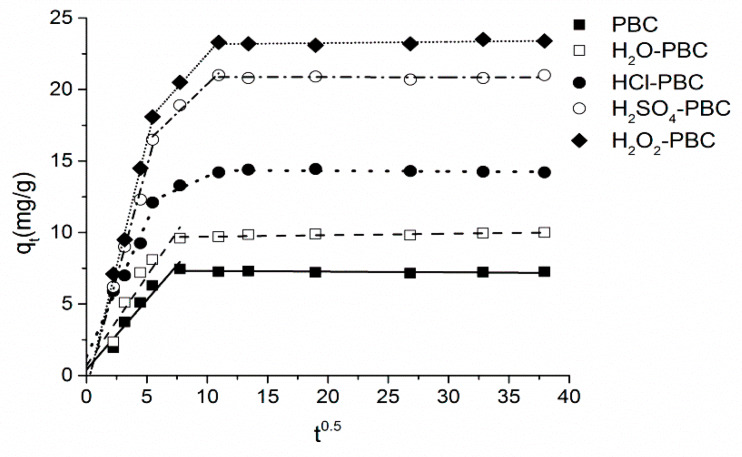
Adsorption kinetics of NH_4_^+^-N on the biochar samples as well as modeling through the intraparticle diffusion model (t is the adsorption time).

**Table 1 materials-13-02270-t001:** Physiochemical properties of the original and acid-modified biochar.

Samples	Ash Content	Volatile Content	Water Content	Fixed Carbon	pH_pzc_
	wt.%	wt.%	wt.%	wt.%	pH Unit
PBC	32.54	24.36	0.74	42.36	9.20
H_2_O-PBC	11.76	30.30	0.98	56.96	8.60
HCl-PBC	8.52	27.77	0.78	62.93	5.00
H_2_O_2_-PBC	6.83	21.00	0.65	71.52	7.20
H_2_SO4-PBC	7.42	24.88	0.82	66.88	6.90

**Table 2 materials-13-02270-t002:** Fitting parameters for the Langmuir and Freundlich isotherms of NH_4_^+^-N adsorption in an aqueous solution on the differently treated biochar.

Adsorbents	Langmuir Model			Freundlich Model		
	Q_m_ (mg·g^−1^)	b (mg·L^−1^)	R^2^	K_F_ (mg·g^−1^)	n	R^2^
PBC	24.58	5.00 × 10^−3^	0.98	0.84	2.10	0.93
H_2_O-PBC	60.50	1.78 × 10^−3^	0.99	0.49	1.56	0.99
HCl-PBC	100.28	4.00 × 10^−3^	0.99	1.89	2.34	0.94
H_2_SO_4_-PBC	120.01	3.12 × 10^−3^	0.99	2.07	1.78	0.95
H_2_O_2_-PBC	123.23	4.40 × 10^−3^	0.99	3.03	1.90	0.95

**Table 3 materials-13-02270-t003:** Kinetic parameters of the pseudo-first-order and pseudo-second-order equations for NH_4_^+^-N adsorption on the five kinds of biochar.

Adsorbents	Pseudo-First-Order Kinetics			Pseudo-Second-Order Kinetics		
	K_1_ (min^−1^)	Q_e_ (mg·g^−1^)	R^2^	v_0_ (g·mg^−1^·min^−1^)	Q_e_ (mg·g^−1^)	R^2^
PBC	0.15	7.20	0.99	0.79	7.53	0.92
H_2_O-PBC	0.15	9.80	0.99	1.03	10.29	0.96
HCl-PBC	0.16	14.22	0.95	1.69	14.83	0.94
H_2_SO_4_-PBC	0.12	20.65	0.98	1.75	21.65	0.96
H_2_O_2_-PBC	0.12	22.93	0.99	1.98	24.10	0.98

**Table 4 materials-13-02270-t004:** Comparison of the maximum ammonium adsorption capacity onto various adsorbents.

Adsorbent	Capacity NH_4_^+^-N (mg·g^−1^)	Concentration Range NH_4_^+^-N (mg·L^−1^)	Contact Time	pH	Temperature (°C)	Ref.
PBC	24.58	20–1000	24 h	≈7	25	This article
H_2_O-PBC	60.50	20–1000	24 h	≈7	25	This article
HCl-PBC	100.28	20–1000	24 h	≈7	25	This article
H_2_SO_4_-PBC	120.01	20–1000	24 h	≈7	25	This article
H_2_O_2_-PBC	123.23	20–1000	24 h	≈7	25	This article
Peanut shell biochar (PS)	243.30	10–500	5–10 h	≈7	25–50	Gao et al., 2015
NaOH modified PS (mPS)	313.90	10–500	5–10 h	≈7	25–50	Liu et al., 2016
Maple wood biochar (MW)	0.46–0.87	0–100	16 h	-	-	Wang et al., 2016
H_2_O_2_ oxidized MW	1.35–7.23	0–100	16 h	-	-	Wang et al., 2016
Rice husk biochar	39.80	250–1400	96 h	5~8	25–45	Kizito et al., 2014
Poulty litter biochar (water washed)	1.33	0–10	24 h	≈7	-	Tian et al., 2016
Bentonite hydrochar	23.67	200	25 h	6	30	Ismadji et al., 2016

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
