# Peer review of "Characterization of Acid-Aged Biochar and Its Ammonium Adsorption in an Aqueous Solution"

_materials, 2020, doi:10.3390/ma13102270_

Round 1

Reviewer 1 Report

Abstract

Line 14 “in this study….” Rewrite sentence for clarity

Please include some of the quantitative results in the abstract

Intro

Line 25 for not of

Line 25 space around comma

Line 26 area to areas

Line 27 vegetable to vegetables

Line 35 remove “with a large amount”

Line 36 change “accelerating the acidification of the soil” to “accelerating soil acidification”

Line 39-40 last sentence please rewrite for clarity

Paragraph 2 – it is important to discuss the biochar characteristics and mechanisms that have found to be related to ammonium uptake (e.g. surface area) and the factors that drive these such as feedstock, production temperature and time, etc.

Line 54 the word hardly does not make sense here

Line 61 remove the word “and”

Line 64 during the application? Do you mean it would lose its beneficial properties over time?

Methods

Line 89 at what temperature and how long were they dried? Did you measure the moisture content before production?

Line 93 how did you measure the pH? Was it added to DI water? We have found typical BET measurements to be ineffective with biochar, were you able to verify a procedure to obtain accurate results? Why did you not measure the oxygen content or other characteristics especially if you were planning to artificially oxidize?

Was just soaking the biochars enough to modify the biochars? Can you please cite the research that indicates this is a suitable method for modification?

Line 130 do not start a sentence with a number unless you write it out

Line 133 what does taken into account mean? Were they controlled and what were they controlled to be? Please be more specific

Results

Line 204 how do you explain the increase in fixed carbon?

Line 205 did the ask content just decrease because it was lost in the solution?

Line 210 doesn’t this mean that your BET evaluation may have been flawed? The change in procedure needs to be explain in the methods not the results

Figure 1 I don’t understand how you measured and determined pore diameter

Figure 1 caption is needs to be corrected

Line 247 remove the word “were”

Line 260 change charcoal to biochar

Line 316 I think the issue may be the method of oxidization as mentioned in the methods section

Line 371 is this just a charge related thing? Also what are the corresponding pH’s you never mentioned if you controlled these?

Line 403 I am concerned about the conclusions you draw from SEM and BET analysis. You need to explain how the BET was done in a way that can be verified to draw conclusions from the data. Also, I would like more discussion on how the two measurements support your conclusion.

Conclusions

Line 414 change compare to compared

Line 416 you need to substantiate that these processes produced biochars that were similar to aged biochars in the results but comparing the measurements of the biochar characteristics to literature changes in biochar aged in soils, this is a really critical part of your justification that is missing in the text

Line 418 I would like some more discussion in the previous section on the maximum sorption as compared to other studies

Author Response

Dear Reviewers, 

We would like to express our sincere appreciation for you careful reading and invaluable comments to prove this paper. We have addressed all the issues you raised and the amendments made are mentioned below with reference to appropriate paragraphs and sections of the revision manuscript.  

  1. Abstract: Line 14 “in this study….” Rewrite sentence for clarity
  2. Please include some of the quantitative results in the abstract
  3. Intro: Line 25 for not of
  4. Line 25 space around comma
  5. Line 26 area to areas
  6. Line 27 vegetable to vegetables
  7. Line 35 remove “with a large amount”
  8. Line 36 change “accelerating the acidification of the soil” to “accelerating soil acidification”
  9. Line 39-40 last sentence please rewrite for clarity
  10. Paragraph 2 – it is important to discuss the biochar characteristics and mechanisms that have found to be related to ammonium uptake (e.g. surface area) and the factors that drive these such as feedstock, production temperature and time, etc.
  11. Line 54 the word hardly does not make sense here
  12. Line 61 remove the word “and”
  13. Line 64 during the application? Do you mean it would lose its beneficial properties over time?

All the spelling and grammar mistakes have been corrected and the inappropriate description has been deleted in the revised paragraph.

  1. Methods: Line 89 at what temperature and how long were they dried? Did you measure the moisture content before production?

We dried the peanut shell in oven at 80℃,we did not measure the moisture content of peanut shell but we dried it till it reach the constant weight.

  1. Line 93 how did you measure the pH? Was it added to DI water? We have found typical BET measurements to be ineffective with biochar, were you able to verify a procedure to obtain accurate results? Why did you not measure the oxygen content or other characteristics especially if you were planning to artificially oxidize?

We measure the pH of biochar follow the IBI (International Biochar Initiative) recommendation, biochar: DI water is 1:20, and were shaken as 150rpm for 1.5h [1]. The BET measurements were talking in the blow section. And the oxygen content were measured through elemental composition analysis and we also measure the element composition especially O content on the surface of biochar through semiquantitative analysis (XPS and EDS). We have also found that typical BET measurements were not clear enough for biochar textural properties identify, when the biochar has too much ash content. Our solution is to degas the material as long as we can in low temperature to reduce the ash impact, and separate the pore structure analysis into two parts, micropore and other pore structure.  

  1. Was just soaking the biochars enough to modify the biochars? Can you please cite the research that indicates this is a suitable method for modification?

The aging process of biochar is through the oxidation process on its surface, in the current experimental research, there are many ways of aging biochar, such as physical aging[2], chemical aging [3, 4], and biological aging. In this research we used chemical aging method to obtain aged biochar in acid and oxidized conditions.

  1. Line 130 do not start a sentence with a number unless you write it out

The amendment has been completed in the paragraph.

  1. Line 133 what does taken into account mean? Were they controlled and what were they controlled to be? Please be more specific

 This sentence is incorrect and had been deleted.

  1. Results: Line 204 how do you explain the increase in fixed carbon?

Fix carbon(%) = 1- ash content(%)- water content(%)- volatile content(%), the increase of dix carbon of biochar shows the increase of PBC stability after acid aging, and was related to the decrease of ash content of biochar.

  1. Line 205 did the ash content just decrease because it was lost in the solution?

We think most of ash content can be removed in the solution, and a small part is removed during oxidization process of aging.

  1. Line 210 doesn’t this mean that your BET evaluation may have been flawed? The change in procedure needs to be explain in the methods not the results

This part is moved to the method part, we do both CO2 an N2 method to decrease the enhance the accuracy of BET evaluation.

  1. Figure 1 I don’t understand how you measured and determined pore diameter

Figure 1 A, B is the CO2 and N2 adsorption and desorption isotherm curves, and C,E and D,F is the pore size distribution curve, we use pore diameter as abscissa.

  1. Figure 1 caption is needs to be corrected
  2. Line 247 remove the word “were”
  3. Line 260 change charcoal to biochar

The amendments have been completed in the revision manuscript.

  1. Line 316 I think the issue may be the method of oxidization as mentioned in the methods section

In this research we mentioned both acid and oxidized aging method, and here the oxidization method means the H2O2 treatment.

  1. Line 371 is this just a charge related thing? Also what are the corresponding pH’s you never mentioned if you controlled these?

We use the kinetics models to identified the diffusion form of aged biochar adsorption behavior, it is not relative to charge, but in this experiments we do set the pH value as 7±0.5 for the batch sorption experiments to obtain the best absorbent effect.  

  1. Line 403 I am concerned about the conclusions you draw from SEM and BET analysis. You need to explain how the BET was done in a way that can be verified to draw conclusions from the data. Also, I would like more discussion on how the two measurements support your conclusion.

We have explained more about the BET results at line 203, and give more details in the method part. The SEM and EDS semiquantitative analyzed elements and identified the elements distribution on the surface of biochar through mapping.

  1. Conclusions: Line 414 change compare to compared

The amendment has been completed in the revised paragraph.

  1. Line 416 you need to substantiate that these processes produced biochars that were similar to aged biochars in the results but comparing the measurements of the biochar characteristics to literature changes in biochar aged in soils, this is a really critical part of your justification that is missing in the text

The explanation has been added in section 2.1, in this article we used acid incubation to simulate the acidic and oxidized process in acid farmland, which is just an ideal condition for the mechanism analysis, and we have some further real site and long-term research for aging biochar.  

  1. Line 418 I would like some more discussion in the previous section on the maximum sorption as compared to other studies

The discussion about maximum sorption has added in section 3.2.

Reference

[1].  Buss, W.; Jansson, S., et al., Unexplored potential of novel biochar-ash composites for use as organo-mineral fertilizers. J Clean Prod 2019, 208, 960-967.

[2].  Shi, K.; Xie, Y., et al., Natural oxidation of a temperature series of biochars: Opposite effect on the sorption of aromatic cationic herbicides. Ecotox Environ Safe 2015, 114, 102-108.

[3].  Hale, S.; Hanley, K., et al., Effects of Chemical, Biological, and Physical Aging As Well As Soil Addition on the Sorption of Pyrene to Activated Carbon and Biochar. Environ Sci Technol 2011, 45, (24), 10445-10453.

[4].  Qian, L.; Chen, B., Interactions of Aluminum with Biochars and Oxidized Biochars: Implications for the Biochar Aging Process. Journal of Agricultural and Food Chemistry 2014, 62, (2), 373-380.

Reviewer 2 Report

The paper submitted by Wang et al. is devoted to ammonium removal from aqueous solution onto biochar artificially aged by chemical treatment. The topic is not new as biochar ageing and its testing as a sorbent of different pollutants, including ammonium has been evidenced in the literature.

However, this paper is prepared and described properly. The biochar before and after modification is characterized in details, which is valuable. The authors made an effort to explain the mechanisms of ammonium removal with the tested biochar. Therefore, after minor revision, I recommend accepting this paper and publish in the Materials.

Please see some specific comments below:

  1. Please rephrase the sentence in L78, p2. What is 'normal soil remediation'? Biochar is not soil remediation in itself but is an amendment used for soil remediation.
  2. Please change the objectives of this paper (L83-84, p2). This is because it looks as biochar was applied for a long-term in farmland. However, such tests have not been performed in this study.
  3. Please check the data '20 to 1000 mg/L' (L140 p3) vs. '20-800 mg/L' (L131 p3). Which data is correct?
  4. Please rephrase sentence at L140 p3.
  5. Some spelling mistakes are present in the text, e.g. 'distill', 'adsoprtion' etc. Please check.
  6. Please explain the 'TC' abbreviation (L132 p3).
  7. 'Charcoal' is not the same as 'biochar' (L260 p7). Please unify.
  8. Discussion can be improved. Could you compare the adsorption capacities of biochar for ammonium with the results obtained by other authors?

Author Response

Dear Reviewers, 

We would like to express our sincere appreciation for you careful reading and invaluable comments to prove this paper. We have addressed all the issues you raised and the amendments made are mentioned below with reference to appropriate paragraphs and sections of the revision manuscript.

  1. Please rephrase the sentence in L78, p2. What is 'normal soil remediation'? Biochar is not soil remediation in itself but is an amendment used for soil remediation.

The modification has been completed in the revision manuscript.

  1. Please change the objectives of this paper (L83-84, p2). This is because it looks as biochar was applied for a long-term in farmland. However, such tests have not been performed in this study.

The modification has been completed in the revision manuscript. We set up the aging process according the acid conditions of long-term farmland, further in site research are in progress.

  1. Please check the data '20 to 1000 mg/L' (L140 p3) vs. '20-800 mg/L' (L131 p3). Which data is correct?
  2. Please rephrase sentence at L140 p3.

We used 1000mgN/L (NH4)2SO4 as the stock solution, and for the adsorption experiments we used the concentrations from 20-1000mgN/L, which made through thinning the stock solution to 20-800mgN/L. We have changed the misunderstanding part in this section.

  1. Some spelling mistakes are present in the text, e.g. 'distill', 'adsoprtion' etc. Please check.

The spelling error has been corrected in the revision manuscript.

  1. Please explain the 'TC' abbreviation (L132 p3).

This sentence is incorrect and had been deleted. (“TC” was a clerical error).

  1. 'Charcoal' is not the same as 'biochar' (L260 p7). Please unify.

The modification has been completed in the revision manuscript.

  1. Discussion can be improved. Could you compare the adsorption capacities of biochar for ammonium with the results obtained by other authors?

The discussion about maximum sorption has added in section 3.2.

Round 2

Reviewer 1 Report

THANK YOU FOR YOUR RESPONSE, REMAINING ISSUES ARE BELOW.

Line 14 “in this study….” Rewrite sentence for clarity - THIS HAS NOT BEEN CLARIFIED, THIS WAS NOT A FARMLAND STUDY PLEASE REVISE

Line 18 - significant digits here and elsewhere in the paper

Line 22 - remove "of" there are a few other English errors in the abstract now that need to be addressed

Intro

Line 39-40 THIS IS NOW LINE 44-67, STILL NOT CLEAR last sentence please rewrite for clarity

Paragraph 2 – it is important to discuss the biochar characteristics and mechanisms that have found to be related to ammonium uptake (e.g. surface area) and the factors that drive these such as feedstock, production temperature and time, etc.YOUR ENTIRE PAPER IS ABOUT AMMONIUM BINDING TO BIOCHAR, YOU DID NOT ADD ANYTHING TO THE INTRO ABOUT THIS, YOU CUT AND THEN PASTED THE SAME PARAGRAPH. THERE IS LOTS OF LITERATURE THAT NEEDS TO BE CITED HERE ON THE RESEARCH ON MECHANISMS OF BIOCHAR NH4 UPTAKE, I WILL NOT RECOMMEND TO PUBLISH UNTIL THIS IS COMPLETED PLEASE.

Methods

Line 89 at what temperature and how long were they dried? Did you measure the moisture content before production? IT SHOULD SAY "UNTIL IT REACHED" I DONT UNDERSTAND YOUR RESPONSE. HOW DID YOU KNOW IT WAS A CONSISTENT WEIGHT UNLESS YOU MEASURED IT?

Line 93 how did you measure the pH? Was it added to DI water? I DO NOT SEE YOUR RESPONSE FOR PH METHODS ADDED TO THE PAPER.

Was just soaking the biochars enough to modify the biochars? Can you please cite the research that indicates this is a suitable method for modification? I DO NOT NECCESARILY AGREE WITH THIS RESPONSE BUT I WILL NOT HOLD THE PAPER UP BECAUSE OF IT, BUT THERE ARE MANY ACCEPTED AND PUBLISHED METHODS FOR OXIDATION, EVEN PARTICULARLY FOR BIOCHAR, I AM NOT SURE WHY YOU DIDN'T USE THOSE

Results

Line 205 did the asH content just decrease because it was lost in the solution? I DO NOT SEE WHERE YOU ADDED THIS TO THE PAPER

Line 316 I think the issue may be the method of oxidization as mentioned in the methods section I DO NOT THINK YOU UNDERSTOOD MY COMMENT. I THINK THIS SHOWS THAT NOT USING AN ACCEPTED METHOD FOR OXIDATION WAS FLAWED AS IT DID NOT WORK VERY WELL, THEN YOU SAY IN THE CONCLUSIONS IT CAN, I DO NOT FIND THIS A LOGICAL CONCLUSION

Conclusions

Line 416 you need to substantiate that these processes produced biochars that were similar to aged biochars in the results but comparing the measurements of the biochar characteristics to literature changes in biochar aged in soils, this is a really critical part of your justification that is missing in the text I DO NOT SEE A DISCUSSION ADDITION THAT SHOWS YOUR BIOCHAR CHARACTERISTICS WERE SIMILAR TO REPORTED STUDIES OF SOIL AGING

Line 418 I would like some more discussion in the previous section on the maximum sorption as compared to other studies I DO NOT SEE THIS INCLUDED

Author Response

Dear Reviewers, 

We would like to express our sincere appreciation again for you careful reading and invaluable comments for this paper. We have addressed all the issues you raised and the amendments made are mentioned below with reference to appropriate paragraphs and sections of the revision manuscript.  

  1. Line 14 “in this study….” Rewrite sentence for clarity - THIS HAS NOT BEEN CLARIFIED, THIS WAS NOT A FARMLAND STUDY PLEASE REVISE

  1. Line 18 - significant digits here and elsewhere in the paper

  1. Line 22 - remove "of" there are a few other English errors in the abstract now that need to be addressed

Author reply: Thank you for your good suggestion. All the spelling and grammar mistakes have been corrected and the inappropriate description has been deleted in the revised paragraph.

Intro

  1. Line 39-40 THIS IS NOW LINE 44-67, STILL NOT CLEAR last sentence please rewrite for clarity

  1. Paragraph 2 – it is important to discuss the biochar characteristics and mechanisms that have found to be related to ammonium uptake (e.g. surface area) and the factors that drive these such as feedstock, production temperature and time, etc.YOUR ENTIRE PAPER IS ABOUT AMMONIUM BINDING TO BIOCHAR, YOU DID NOT ADD ANYTHING TO THE INTRO ABOUT THIS, YOU CUT AND THEN PASTED THE SAME PARAGRAPH. THERE IS LOTS OF LITERATURE THAT NEEDS TO BE CITED HERE ON THE RESEARCH ON MECHANISMS OF BIOCHAR NH4 UPTAKE, I WILL NOT RECOMMEND TO PUBLISH UNTIL THIS IS COMPLETED PLEASE.

Author reply: Thank you for your good suggestion. We rewrite the sentence and paragraph 2 according to your useful advice (line 84).

Methods

  1. Line 89 at what temperature and how long were they dried? Did you measure the moisture content before production? IT SHOULD SAY "UNTIL IT REACHED" I DONT UNDERSTAND YOUR RESPONSE. HOW DID YOU KNOW IT WAS A CONSISTENT WEIGHT UNLESS YOU MEASURED IT?

Author reply: Thank you for your good suggestion. We have corrected our grammar mistake follow your advice and sorry for our unclear explanation. Here “dying until it reached the constant weight” means that we do measure the weight of biochar before we put it in the oven, and we keep dying to achieve constant weight by measuring the weight after dying (line 209).

  1. Line 93 how did you measure the pH? Was it added to DI water? I DO NOT SEE YOUR RESPONSE FOR PH METHODS ADDED TO THE PAPER.

Author reply: Thank you for your good suggestion and we are sorry for our unclear explanation, we have added more detail in the paragraph (line 213).

  1. Was just soaking the biochars enough to modify the biochars? Can you please cite the research that indicates this is a suitable method for modification? I DO NOT NECCESARILY AGREE WITH THIS RESPONSE BUT I WILL NOT HOLD THE PAPER UP BECAUSE OF IT, BUT THERE ARE MANY ACCEPTED AND PUBLISHED METHODS FOR OXIDATION, EVEN PARTICULARLY FOR BIOCHAR, I AM NOT SURE WHY YOU DIDN'T USE THOSE

Author reply: Thank you for your useful advice, we considered s. There are many oxidizing methods for biochar modification such as KOH, H2O2, oxidizing acid (HNO3/H2SO4) or mixed oxidizing agents [1-3], etc. However, the aging process under aging conditions sometimes will not as similar as the active process of biochar. The general modified process normally include modifying agent treatment and 2nd pyrolysis process in order to gain more porous structure and surface area of biochar . But in aging process, the temperature during the whole process cannot reach as high as pyrolysis. Therefore in this article, we trying to find out whether the adsorption ability of biochar will change under acid and oxidizing conditions, we built up a simulation experiments to compare the sorption ability before and after acid treatments. And because of the flaws of our research, we are now doing further research of biochar adsorption capacity under natural aging conditions.

Results

  1. Line 205 did the asH content just decrease because it was lost in the solution? I DO NOT SEE WHERE YOU ADDED THIS TO THE PAPER

Author reply: Thank you for your good suggestion. We add more illustration about ash content at line 244.

  1. Line 316 I think the issue may be the method of oxidization as mentioned in the methods section I DO NOT THINK YOU UNDERSTOOD MY COMMENT. I THINK THIS SHOWS THAT NOT USING AN ACCEPTED METHOD FOR OXIDATION WAS FLAWED AS IT DID NOT WORK VERY WELL, THEN YOU SAY IN THE CONCLUSIONS IT CAN, I DO NOT FIND THIS A LOGICAL CONCLUSION

Author reply: Thank you for your good suggestion. We give more explanation about the oxidizing aged method we chose at question 8, and we change the conclusion to a more logical statement.

Conclusions

  1. Line 416 you need to substantiate that these processes produced biochars that were similar to aged biochars in the results but comparing the measurements of the biochar characteristics to literature changes in biochar aged in soils, this is a really critical part of your justification that is missing in the text I DO NOT SEE A DISCUSSION ADDITION THAT SHOWS YOUR BIOCHAR CHARACTERISTICS WERE SIMILAR TO REPORTED STUDIES OF SOIL AGING
  2. Line 418 I would like some more discussion in the previous section on the maximum sorption as compared to other studies I DO NOT SEE THIS INCLUDED

Author reply: Thank you for you valuable advice, we rewrite the conclusion part referred to your advice. Although literatures have proved that this method can simulate the aging behavior of biochar hundreds of years, but biochar applied into the environment by aging factor is various, such as acid rain on the friction on the physical, chemical, humus in the soil, environmental temperature, humidity, biological various microbes and so on, these factors often leads to aging effects on biological carbon adsorption is not exactly the same. This experiment is limited by the laboratory conditions. In order to comprehensively investigate the environmental effects of biochar, long-term cultivation experiments of biochar soil should be conducted to comprehensively consider the influence of various epigenic geochemical factors. And the discussion about maximum adsorption we have added at line 545.

[1].  Fan, Q.; Cui, L., et al., Effects of Wet Oxidation Process on Biochar Surface in Acid and Alkaline Soil Environments. Materials 2018, 11, (12), 2362.

[2].  Cross, A.; Sohi, S. P., A method for screening the relative long-term stability of biochar. Gcb Bioenergy 2013, 5, (2), 215-220.

[3].  Qian, L.; Chen, B., Interactions of Aluminum with Biochars and Oxidized Biochars: Implications for the Biochar Aging Process. Journal of Agricultural and Food Chemistry 2014, 62, (2), 373-380.

Round 3

Reviewer 1 Report

The literature review is still very broad and should focus more on past studies of ammonium/ammonia biochar.

I do not see in the discussion a paragraph that discusses the diffference int he characteristics of the oxidized biochar versus soil aged biochar as that seems to be a big focus of your reason for doing the work. 

Author Response

Dear Reviewers, 

We would like to express our sincere appreciation again for you careful reading and invaluable comments for this paper. We have addressed all the issues you raised and the amendments made are mentioned below with reference to appropriate paragraphs and sections of the revision manuscript.

Reviewer: The literature review is still very broad and should focus more on past studies of ammonium/ammonia biochar.

I do not see in the discussion a paragraph that discusses the diffference int he characteristics of the oxidized biochar versus soil aged biochar as that seems to be a big focus of your reason for doing the work. 

Author reply: We add more introductions at line 73 and we add discussions about the comparison of ammonium adsorption  performance between our research and other studies at line 1413 and Table 5.

The English language editing of our article can be seen in revised mode and the changes of the title and abstract were highlight in red.